# Histopathological Biocompatibility Evaluation of TheraCal PT, NeoMTA, and MTA Angelus in a Murine Model

**DOI:** 10.3390/jfb14040202

**Published:** 2023-04-06

**Authors:** Francelia Quiñonez-Ruvalcaba, Carlos Bermúdez-Jiménez, Luis Alejandro Aguilera-Galavíz, Francisco G Villanueva-Sánchez, Salvador García-Cruz, César Gaitán-Fonseca

**Affiliations:** 1Programa de “Especialidad en Odontopediatría, Unidad Académica de Odontología, Universidad Autónoma de Zacatecas”, Zacatecas 98000, Mexico; 2Unidad Académica de Odontología, Universidad Autónoma de Zacatecas “Francisco García Salinas”, Zacatecas 98000, Mexico; 3Interdisciplinary Research Laboratory, Oral and Maxillofacial Pathology Area, Universidad Nacional Autónoma de México, Escuela Nacional de Estudios Superiores Unidad León, León 37684, Mexico; 4Facultad de Medicina Humana y Ciencias de la Salud, Universidad Autónoma de Zacatecas, Zacatecas 98000, Mexico

**Keywords:** cellular inflammatory infiltrate, pulp tissue disorganization, reparative dentin, pulpotomy, murine model

## Abstract

The aim of this study was to evaluate the biocompatibility of the regeneration of the dentin–pulp complex in a murine model with different treatments with MTA Angelus, NeoMTA, and TheraCal PT. An in vivo controlled experimental study of 15 male Wistar rats forming three study groups, the upper and lower central incisors were selected where pulpotomies were conducted, leaving a central incisor as control at 15, 30, and 45 days. For data analysis, these were expressed as mean ± standard deviation and were examined by Kruskal–Wallis test. Three factors were analyzed as follows: “inflammatory infiltrate; disorganization of pulp tissue, and the formation of reparative dentin”. No statistical significance was found between the different groups (*p >* 0.05). Treatment with these three biomaterials (MTA, TheraCal PT, and Neo MTA) presented an inflammatory infiltrate and slight disorganization of the odontoblast layer in the pulp tissue of a murine model, with normal coronary pulp tissue and the formation of reparative dentin in the three experimental groups. Thus, we are able to conclude that all three are biocompatible materials.

## 1. Introduction

During the growth of the dentin–pulp complex, the pulp cells produce dentin, nerves, and blood vessels; dentin would not exist were it not produced by the odontoblasts, and dental pulp depends on the protection offered by the dentin and the enamel [1,2,3]. Similarly, the dynamic comprising the dentin–pulp complex implies that the impacts of the dentin can alter the pulp components, and the alterations of the pulp can, in turn, alter the quality and amount of dentin produced. These reactions are carried out with changes in the fibroblasts, nerves, blood vessels, odontoblasts, leukocytes, and in the immune system [1,2,3,4,5,6,7].

Pulpotomy is a conservative treatment that consists of the removal inflamed vital pulp in the pulp chamber, followed by the medicines for the stimulation and repair of the remainder of the vital pulp. Its main objective is that the radicular tissues found to be clinically healthy continue to develop physiologically with the natural process [2,8,9,10,11].

In pulpotomies, there are three principal focuses; devitalization: in this, vital tissue is destroyed; preservation: here, it is sought to preserve pulp vitality to the maximal degree without inciting dentin reparation; and regeneration: attempts to stimulate the pulp function to provoke the formation of a dentinal bridge, that is, reparative induction [12,13].

Among the agents most employed for the development of pulpotomies, we should mention calcium hydroxide. This is considered as the medicine of choice in direct and indirect pulp protection and vital pulpotomy. It is produced through the mixture of calcium oxide (lime) with water [14]. It favors the formation of reparative dentin, is biocompatible and possesses antimicrobial properties, a pH of 12.4, with the capacity to form reparative dentin and mineralized tissue. Some of its disadvantages include that it has high solubility and a lack of adhesion to dental tissues; as well as that, like a restorative material, it can give rise to filtrations [15]. Its use frequently causes the development of chronic inflammation of the pulp and the incidence of radicular reabsorption. Could hypothetically cause internal reabsorption, stimulated by material and produced by a coagulum that intervenes between the material and pulp [16].

Mineral trioxide aggregate (MTA), a compound employed to seal the communications between the conduit system and the periodontium, due to its marginal sealing properties, a pH of 12.5, and antibacterial activity [17], presents certain advantages in terms of its application in the pulpotomy treatments of temporary molars, in that this material does not produce signs or symptoms of pulp pathology and preserves healthy radicular pulp, in addition to not being toxic to the tissues [16].

Currently, NeoMTA NuSmile, which has been developed for pediatric odontology, has a powder–liquid presentation composed of tricalcium silicate, bismuth oxide, tricalcium aluminate, calcium sulfate, and plaster of Paris. The liquid is composed of a water-based gel with thickening agents and water-soluble polymers. NeoMTA NuSmile does not stain or discolor the tooth, is water-resistant, and has a rapid setting time [18].

TheraCal PT is a double-cured and resin-modified calcium silicate; it is alkaline, biocompatible, and a calcium releaser; it is indicated for use in the vital pulp; it protects the dental pulp complex, and is favorable for successful pulp therapy for pediatric patients. It possesses physical properties to reinforce the primary-dentition teeth, which alleviates the need for total coverage of the treated dentition. The use of a long-lasting dicalcium or tricalcium silicate-modified hydrophilic resin enables the performance of more conservative restorations [19].

Deriving as it does from the manufacture of this entire gamut of biomaterials, it is always important to know the physical–chemical characteristics in a clinical environment. For this, in vivo models permit us to obtain the most approximate information of the behavior that can occur in a real clinical scenario [20,21].

For medical and experimental odontology, the mouse is a model organism that offers many advantages, such as those employed in biomedical investigations, above all in physiology, immunology, toxicology, oncology, pharmacology, and in the study of behavior with respect to other genetic models such as the *Drosophila* fly [22], the *Caenorhabditis elegans* nematode [22], and even the rat [22].

This latter species shares with humans the privilege of being the most studied mammalian species from the genetic point of view; there are a great number of genetically defined lines, such as consanguine and congenital, in addition to hundreds of mutations and a large number of available chromosomal arrangements [22,23].

Therefore, the aim of this study was to evaluate the biocompatibility of regeneration of the dentin–pulp complex in a murine model with different pulpotomy treatments (MTA Angelus, NeoMTA, and TheraCal PT) at 15, 30, and 45 days. 

## 2. Materials and Methods

The study was approved by the Investigation Commission (Registry no. CI-UAE-08–2021) and Ethics Committee in Investigation (Registry no: 026/CEI-UAE-UAZ/2021) of the Nursing Academic Unit, Autonomous University of Zacatecas (UAZ). It was a controlled in vivo experimental study using a split-mouth assay during the period comprising May through August 2021, with a study universe of 15 male Wistar rats, 13–14 weeks of age, and with an approximate weight of 250–300 g each, from which three study groups were formed (“G1 MTA Angelus”, “G2 NeoMTA NuSmile”, and “G3 TheraCal PT^”^) (Table 1), made up of five rats each. The upper and lower central incisors were selected, obtaining a total of 60 dental organs and evaluated in three time periods: 15, 30, and 45 days.

The pulpotomy treatment was carried out with the following organization: MTA (right upper central incisor); TheraCal PT (left upper central incisor); NeoMTA (left lower central incisor), and negative control (right lower incisor), access to the pulp chamber without obturation. For carrying out the pulpotomy, general anesthesia was performed with Sevoflurane (FHADOTIL, Sevoflurane 100%, 250-mL flask; Farmacéutica Hispanoamericana), within a hermetic recipient for a time of 3 min; after this, a mouth-opening retractor) designed with 0.040”-caliber wire (American Orthodontics) was employed. The cavity was prepared with a #1 mm carbide ball drill, the pulp chamber was removed, and washing with sterile water (PiSA®FARMACEUTICA, Jalisco, México). Hemorrhage was controlled through the pressure technique and placement of the materials was according to the manufacturer specifications. All cavities of the study groups were coronally sealed with Ketac Universal. Once the different evolution times had elapsed, the samples of all of the groups were submitted to euthanasia by means of sedation with the use of Sevoflurane for between 5 and 10 min. A maxillary and mandibular dissection was conducted in each rat, and this was placed in a hermetic recipient with formol 10% (Formol 37%; Drogas Tacuba, S.A. de C.V., CDMX, México). The samples were sent for histopathologic study to the Laboratory of Interdisciplinary Investigation, ENES, UNAM, Campus León, Mexico.

For the histopathologic procedure, the samples were submitted to the following stages: (a) measuring sample; (b) decalcification with nitric acid 5%; (c) slicing of the samples; (d) dehydration and clarification in the Histokinette (Leica); (e) embedding in paraffin (Roundfind, Shenyang, China); (f) microtome slicing (Leica); (g) staining with H&E; (h) mounting, and (i) observation with the optic microscope.

As described in the previous stages of the histologic procedure, the pathological samples were extracted from the 10% formol and were placed on a tablet to measure the size of each sample with a Vernier, followed by decalcification with nitric acid 5% in assay tubes, and the samples were set in place. After this, we proceeded to slice the samples on the tablet and, with a Leica-brand low-profile razor, transversal slices were performed at the cervical level, where access for the pulpotomy was realized. Once the slices were obtained, each part was identified, wrapped in filter paper, placed in plastic cassettes for its storage, and deposited in the Leica-brand Histokinette for dehydration and clarification for 12 h. Once this process was terminated, we began the embedding of the samples with paraffin in order to slice them afterward with the microtome into 3 µm slices. Mounting of the slices was carried out on the slides. Hematoxylin & eosin (H&E) staining was performed on the samples due to their simplicity and capacity to allow the visualization of many structures of different tissues. Once this procedure concluded, the samples were observed with the microscope.

For the histological analysis, we conducted a “double-blind study” and a qualitative assessment based on the classification of Table 2, Table 3 and Table 4 according to the modified version of “ISO 10993 and 7405”. The objective of the H&E staining was the evaluation of the inflammatory infiltrate, of the pulp characteristics, and of the formation of the dentin tissue [21]. Stained histological slices were observed with the H&E technique with the microscope at 40×. The photographs were obtained with a Canon Eos Rebel T6 photographic camera of the slowest Ef s 18–55 mm, with the camera coupled to the microscope.

### Statistical Analysis

The data (degrees of regeneration, inflammation, and formation) were expressed as mean ± standard deviation (SD) and were evaluated by Kruskal–Wallis test. The Dunn post hoc test was applied to observe the differences between the different groups. The data were analyzed with the “GraphPad Prism version 8.0.0 statistical software program” (San Diego, CA, USA). A value of *p* < 0.05 was statistically significant.

## 3. Results

It was observed at the histological level that no presence of cellular inflammatory infiltrate (Figure 1) was found, but it was reported that at day 15, NeoMTA demonstrated slightly greater cellular inflammatory infiltrate than MTA Angelus and TheraCal PT, but that, at 30 and 45 days after the utilization of the three materials, the latter were found with grade-1 slightly inflammatory infiltrate. However, statistical significance was not observed (*p* > 0.05) (Figure 2).

With regard to pulp-tissue disorganization (Figure 3), in the negative control the presence of this variable was not observed (15, 30, and 45 days). On the other hand, in MTA Angelus, NeoMTA, and TheraCal PT, there was grade-1 pulp-tissue disorganization. In terms of statistics, statistically significant differences were not observed with a *p* of >0.05 (Figure 4). Likewise, in the formation of reparative dentin (Figure 5), in the negative control there was no presence of this, but in the three materials utilized, a grade-1 result was revealed, with a final result of statistically significant differences not being observed (*p* > 0.05) (Figure 6).

**Figure 1 jfb-14-00202-f001:**
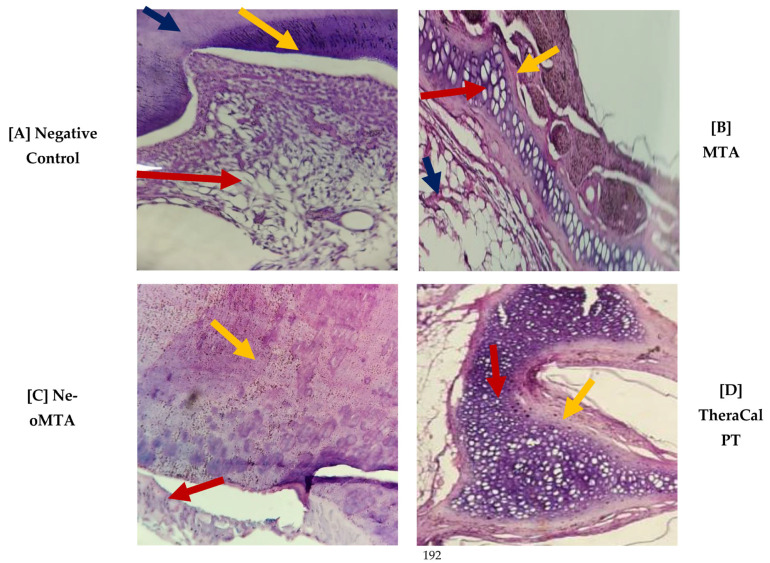
Photomicrograph of Group 1 (15 days), cross-section, stained with H&E. (**A**) at (4×), (**B**–**D**) at (10×).


(A)The stellate epithelium of the dental pulp (red arrow) and the surrounding odontoblastic layer (yellow arrow) and dentin (blue arrow) were observed.(B)The particulate dental material was observed in the form of a cobweb with a basophilic color more purple than the rest of the tissues (red arrow); there was the presence of a pseudocapsule of fibroconnective tissue around it, which was loose and fibrous (yellow arrow); the presence of adipose tissue (blue arrow).(C)The odontoblastic layer (red arrow) was observed, adjacent to the dentin with the presence of integration of the basophilic epithelium that corresponded to the dental material used and that showed us that there was an integration between the dental material and the dentin (yellow arrow).(D)Particulate dental material with a basophilic color that was more purple than the rest of the tissues (red arrow) was observed, and the biology of the mouse encapsulated a pseudocapsule of surrounding fibroconnective tissue that was loosely arranged, fibrous (arrow), yellow); there was the presence of a mild inflammatory infiltrate; it was considered that there was good integration of the material.


**Figure 2 jfb-14-00202-f002:**
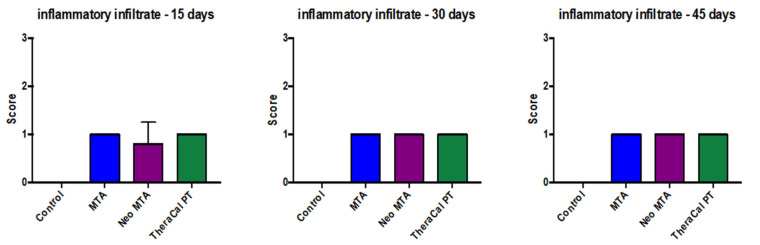
Cellular inflammatory infiltrate in response to the different treatments carried out on rats at 15, 30, and 45 days. No statistical significance was found between the different groups (*p* > 0.05). Each of the experiments was carried out five times.

**Figure 3 jfb-14-00202-f003:**
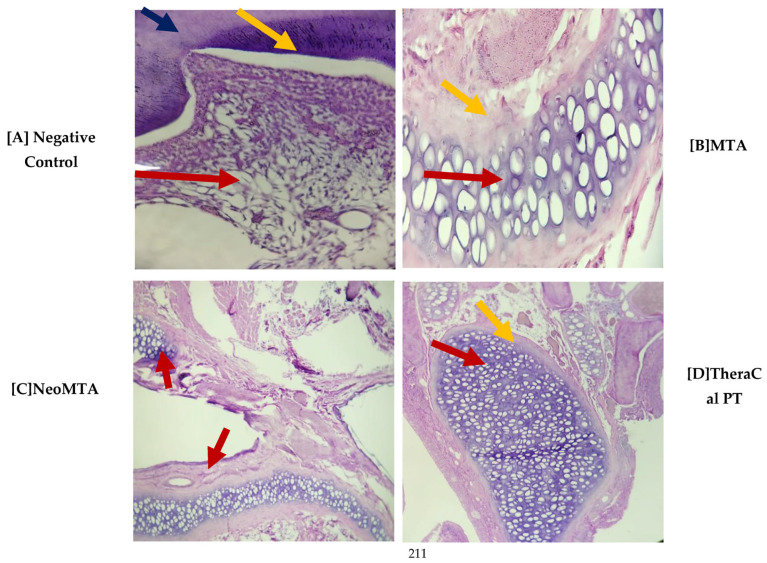
Photomicrograph of Group 2 (30 days) cross-sections stained with H&E. (**A**) at 4×, (**B**) 40×, (**C**,**D**) at 4×.


(A)The stellate epithelium of the dental pulp (red arrow) and the surrounding odontoblastic layer (yellow arrow) and dentin (blue arrow) were observed.(B)A cobweb of basophilic epithelium was observed that was more purple than the rest of the tissues (red arrow), a pseudocapsule of fibroconnective tissue around it that was loose and fibrous (yellow arrow) and adjacent to a mild inflammatory infiltrate; a sign that there was good integration of the material that was used.(C)Two cobweb-shaped basophilic epithelia were observed, which was the dental material that was used (red arrow) and this was encapsulated by a pseudocapsule of fibroconnective tissue (yellow arrow), and adipose tissue was observed (blue arrow).(D)The presence of basophilic epithelium in cobweb mode (red arrow) was observed, with the presence of a pseudocapsule of fibroconnective tissue (yellow arrow).


**Figure 4 jfb-14-00202-f004:**
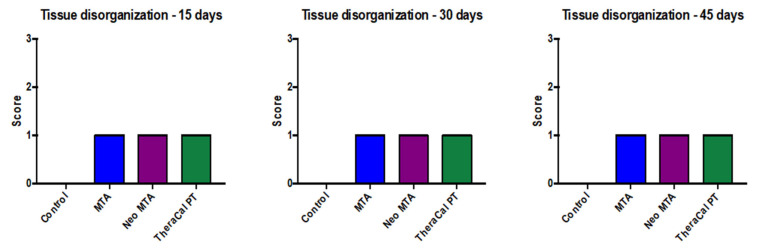
Pulp tissue disorganization in response to the different treatments carried out on rats at 15, 30, and 45 days. No statistical significance was found between the different groups (*p* > 0.05). Each of the experiments was carried out five times.

**Figure 5 jfb-14-00202-f005:**
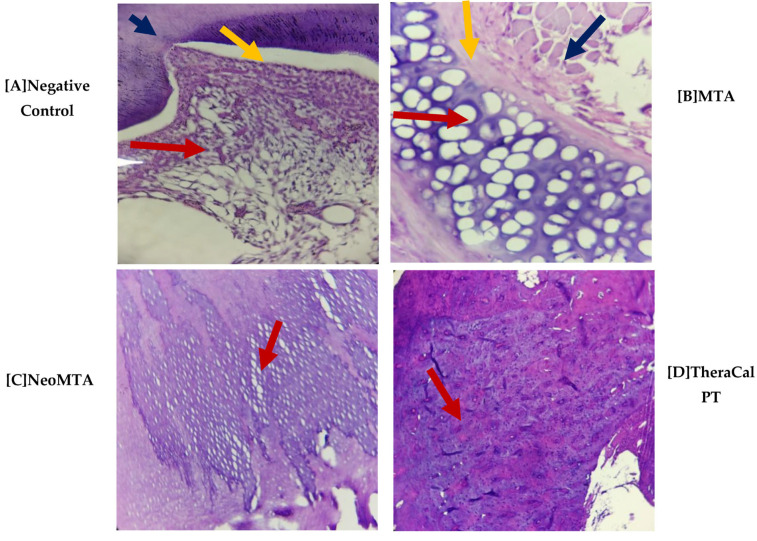
Photomicrograph of Group 3 (45 days) cross-sections stained with H&E. (**A**) at 4×, (**B**) 40×, (**C**) 10× and (**D**) at 40×.


(A)The stellate epithelium of the dental pulp (red arrow) and the surrounding odontoblastic layer (yellow arrow) and dentin (blue arrow) were observed.(B)The particulate dental material was observed in the form of a cobweb with a basophilic color that was more purple than the rest of the tissues (red arrow); there was the presence of a pseudocapsule of surrounding fibroconnective tissue that was loose and fibrous (yellow arrow, and the presence of adipose tissue (blue arrow).(C)Disorganization of the odontoblast layer was observed and adjacent to it there was a cobweb of basophilic epithelium that corresponded to the material and showed us that there was an integration with the dentin (red arrow).(D)Disorganization of the odontoblastic layer was observed and adjacent to it we found the presence of a cobweb of basophilic epithelium that corresponded to the dental material being tested, where there was also an integration with the dentin (red arrow).


**Figure 6 jfb-14-00202-f006:**
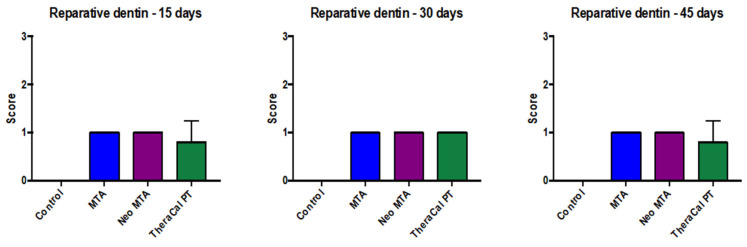
Reparative dentin formation in response to the different treatments carried out on rats at 15, 30 and 45 days. No statistical significance was found between the different groups (*p* > 0.05). Each of the experiments was carried out five times.

With respect to the presence of fibrosis (Table 5), only in the negative control of G1 at 15 days was there the absence of fibrosis; later, at 30 and 45 days, the development of tissue fibrosis was demonstrated. From 15, 30, and 45 days after the employment of MTA Angelus, NeoMTA, and TheraCal PT, there was the presence of tissue fibrosis. On the appearance of tissular necrosis (Table 6), in G1 at 15 days, an absence was observed in the negative control, but in G2 at 30 days and in G3 at 45 days, the presence of tissular necrosis was observed. However, there was, indeed, a presence of tissular necrosis in the three groups at 15, 30, and 45 days, in which MTA Angelus, Neo MTA, and TheraCal PT was employed.

## 4. Discussion

According to the preclinical evaluation of the biocompatibility of the materials for use in dentistry as determined by “NORMA ISO 7405”, it has been established that only non-rodent mammals, such as miniature pigs, dogs and monkeys are adequate for animal investigation. However, in recent years, various biocompatibility studies have been published with the use of rat teeth with the objective of evaluating tissue reactions after pulp exposure, and these dental organs present similarities such as anatomic, histologic, biologic, and physiologic, and are considered as miniatures of human dental organs, considering that the biological reactions of pulp tissue are comparable with those of other mammals [21].

Having as the main objective the evaluation of the regeneration of the dentin–pulp complex that is produced by TheraCal PT, NeoMTA, and MTA in pulpotomies in Wistar rat teeth and observed over a period of 15, 30, and 45 days, to find out if there was a presence of inflammatory infiltrate, pulp-tissue disorganization, and the formation of reparative dentin. In addition, one must bear in mind that rat teeth are a valid model for providing data on the reaction after pulp exposure and that they have the capacity for tissue recovery. In this regard, the use of this model reduced the research sample with ethical and economic advantages [21].

Dental pulp entertains the capacity of recovery under propitious circumstances and, at the tissue level, the former possesses the capacity on its own to diminish the inflammatory infiltrate in the case of the existence of a pulp lesion. In our study, we established a negative-control group in which only pulp exposure was conducted, without the placement of any material. We observed at the histological level that no presence was found of cellular inflammatory infiltrate, but it was reported that, at 15 days, NeoMTA demonstrated a slightly greater inflammatory cell infiltrate than MTA and TheraCal PT, but that at 30 and 45 days after the three materials were employed, these were found at grade 1 of slight inflammatory infiltrate. Nevertheless, no statistical significance was observed (*p* > 0.05).

In terms of pulp-tissue disorganization, in the negative control, the presence was not observed of this variable (15, 30, and 45 days). On the other hand, in MTA, NeoMTA, and TheraCal PT, there was grade-1 pulp tissue disorganization, while statistically significant differences were not observed, with *p* > 0.05. In a similar manner, in the formation of reparative dentin, in the negative control there was no presence of this disorganization, but the three materials utilized yielded grade 1, while in the final result, no statistically significant difference was observed (*p* > 0.05). 

In the presence of fibrosis (Table 4), only in the control group grade 1 at 15 days was there the absence of fibrosis. Later, after 30 and 45 days, the development of tissue fibrosis was demonstrated. At 15, 30, and 45 days after the employment of MTA, NeoMTA, and TheraCal PT, there was the presence of tissular fibrosis. On the appearance of tissular necrosis (Table 5), an absence was observed in the negative control in G1 at 15 days, but in G2 at 30 days and in G3 at 45 days, the presence was observed; however, there was indeed the presence of tissular necrosis in all three groups at 15, 30, and 45 days when MTA, Neo MTA, and TheraCal PT were utilized. 

With respect to the comparison of other results of other, similar studies, Liu & et al. (2015) performed an experimental study in vivo in which the authors utilized Wistar rats, assigning 12 rats for direct pulp capping and eight rats for pulpotomies, in which they used as agents MTA and iRoot BP Plus, a calcium silicate-based bioceramic. The results obtained reported that the iRoot BP Plus could induce a reparative dentin bridge at the site at which the rat dental pulp was exposed mechanically, whether in direct pulp capping or in pulpotomy and that this dentin bridge was stronger than the MTA [24].

In our study, we did not observe any differences in the formation of reparative dentin with NeoMTA and TheraCal PT in comparison with MTA, suggesting that both materials (NeoMTA and TheraCal PT) present similar characteristics in terms of dentin formation. These results were not compatible in that there is, to our knowledge, no previous literature on this. 

On the other hand, Trongkij et al. (2019) observed the activity of a calcium-silicate cement that contained a calcium-chloride accelerator as a capping material in 60 mechanically exposed upper molars of Wistar albino rats, and the following two capping materials were assigned: Bio-MA or white mineral trioxide aggregate (WMTA), over three periods of 1, 7, or 30 days. The authors evaluated the inflammatory cell infiltrate and the formation of reparative dentin. The authors did not observe significant differences in the pulp responses between the two materials. Notwithstanding this, it confirmed that direct pulp capping (DPC) possesses the capacity for auto-reparation in rat dental pulp and differentiation of the pulp cells in calcium silicate-based materials [25]. 

Liu & et al., in 2015, and Trongkij and colleagues, in 2019, conducted studies with calcium silicate-based bioceramic materials in which the results obtained by the authors showed that there was an inflammatory cell infiltrate and the formation of reparative dentin. Therefore, we can infer that all of the medicines employed are biocompatible, without toxic potential, and with regenerative and reparative scenarios, as well as highlighting that TheraCal PT and NeoMTA can be favorable in the treatment of pulpotomy. 

Nonetheless, with regard to cost–benefit comparisons, TheraCal PT has a market price lower than that of NeoMTA from the NuSmile company (Houston, TX, USA); however, NeoMTA has a longer expiration time in comparison with TheraCal PT.

Although there are similarities between the dental organs of rat and human, one must bear in mind that there are limitations: oral media and plaque composition differ between rat and those of the human (normal oral microbiota). In the present work, there was the limitation of conducting the treatments in rat molars, due to the difficult and limited access for placement of the materials. Due to the latter, we were obliged to work with the upper as well as the lower central incisors, by means of sedation with Sevoflurane, and this provided us with a limited work time due to the duration of the effect of the Sevoflurane.

Regarding the histological technique, difficulty was encountered in the performance of histological slices due to the small size of our samples, resulting in complex histological preparation. 

These results were not compatible with those of other studies, due to the scarce information on the biocompatibility of the materials utilized in this investigation on the generation of reparative dentin in in vivo studies, highlighting the importance of our analysis, in that it would be, to our knowledge, the first to report these data in a murine model.

## 5. Conclusions

Treatment with the employed biomaterials (MTA Angelus, TheraCal PT, and NeoMTA) presented an inflammatory infiltrate and slight disorganization of the odontoblast layer in the pulp tissue of a murine model. Normal coronary pulp tissue and the formation of reparative dentin were observed in the three groups. It was concluded that these are materials that are biocompatible with dental tissues and that they exhibit very similar results among themselves. Likewise, conducting randomized clinical studies and cohort studies is recommended for obtaining greater information with respect to biocompatibility in the pediatric area with safer and more reliable information on dental treatments.

## Figures and Tables

**Table 1 jfb-14-00202-t001:** Composition of the commercial materials.

Material	Lot Number	Composition	Manufacturer
MTA Angelus Blanco	102,353	“Tricalcium silicate, dicalcium silicate, tricalcium aluminate, calcium oxide, calcium tungstate, and distilled water”	Angelus
TheraCal PT	2,000,000,454	“Dual-cure resin- modified calcium silicate”	Bisco
NeMTA	2,019,122,801	“Tricalcium and dicalcium silicate and water-based gel”	NuSmile

**Table 2 jfb-14-00202-t002:** “Classification of cellular inflammatory infiltrate” “Adapted with permission from Ref. [21] and ISO 7405:2018”.

Inflammatory Infiltrate	Characterization of Inflammatory Infiltrate
Grade 0	“No inflammatory signs and no presence or with the appearance of a few inflammatory cells in the pulp area corresponding to the exposure zone”
Grade 1	“Slight inflammatory infiltrate with the presence of cells, such as polymorphonuclear leukocytes (PMNL) and mononuclear leukocytes (MNL)”
Grade 2	“Moderate cellular inflammatory infiltrate involving the coronary pulpal tissue”
Grade 3	“Severe cellular inflammatory infiltrate involving the coronary pulp tissue or with abscess characteristics”

**Table 3 jfb-14-00202-t003:** “Classification of pulp tissue disorganization” “Adapted with permission from Ref. [21] and ISO 7405:2018”.

Pulp Tissue Disorganization	Characterization
Grade 0	“Normal tissue“
Grade 1	“Disorganization of the odontoblast layer, but normal coronary pulp tissue”
Grade 2	“Total disorganization of pulp tissue morphology”
Grade 3	“Pulp necrosis“

**Table 4 jfb-14-00202-t004:** “Classification of reparative dentin formation” “Adapted with permission from Ref. [21] and ISO 7405:2018”.

Reparative Dentin Formation	Characterization
Grade 0	“Absence”
Grade 1	“Slight deposition of hard tissue immediately below the exposure zone”
Grade 2	“Moderate hard tissue deposition immediately below exposure zone”
Grade 3	“Intense deposition of hard tissue immediately below the exposure zone, characteristic of a complete dentin bridge”

**Table 5 jfb-14-00202-t005:** Tissue fibrosis.

Period	Control (–)	G1 MTA	G2 NeoMTA	G3 TheraCal PT
15 days	Absent	Present	Present	Present
30 days	Present	Present	Present	Present
45 days	Present	Present	Present	Present

**Table 6 jfb-14-00202-t006:** Tissue necrosis.

Period	Control (–)	G1 MTA	G2 NeoMTA	G3 TheraCal PT
15 days	Absent	Present	Present	Present
30 days	Present	Present	Present	Present
45 days	Present	Present	Present	Present

## Data Availability

All data used in this study are declared in the paper.

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
