# Peer review of "Histopathological Biocompatibility Evaluation of TheraCal PT, NeoMTA, and MTA Angelus in a Murine Model"

_jfb, 2023, doi:10.3390/jfb14040202_

Round 1
Reviewer 1 Report
The study appears to be well-designed and provides valuable information about the biocompatibility and regenerative potential of three different biomaterials for use in dental treatments. The manuscript is of potential interest to the broad readers of J. Funct. Biomater.
However, there can be drawbacks in this research. If the following problems are well-addressed, this reviewer believes that the essential contribution of this paper is important for biocompatible materials.
1. Small sample size of only 15 rats and the fact that it was conducted in a murine model rather than human subjects.
2. The study only evaluated short-term outcomes at 15, 30, and 45 days, so it is unclear whether the observed results would be sustained over a longer period.
3. The study did not compare the efficacy of the three materials in terms of clinical outcomes such as pain relief or the need for additional treatments, which may be important factors in determining their usefulness in dental practice.
Here are some suggestions to improve the research:
1. Sample size: The study included only 15 rats, which may not be sufficient to draw definitive conclusions about the effectiveness of each biomaterial. A larger sample size would increase the statistical power of the study and provide more robust results.
2. Control group: The study used a single control tooth for each rat, which may not be adequate to control for individual variability and other potential confounding factors. A larger control group would improve the validity of the study.
3. Follow-up period: The study evaluated the outcomes at 15, 30, and 45 days after treatment, which may not be sufficient to evaluate the long-term regenerative potential of the biomaterials. Longer follow-up periods would provide more information about the durability and effectiveness of each material over time.
4. Clinical studies: The study suggests conducting randomized clinical studies and cohort studies to obtain more information about the biocompatibility and effectiveness of biomaterials in pediatric dental treatments. Conducting such studies would provide more reliable and generalizable information for clinical practice.
Overall, the study provides important preliminary data about the biocompatibility and regenerative potential of three different biomaterials for use in dental treatments. However, further studies are needed to confirm and expand upon these findings.
Author Response
However, there can be drawbacks in this research. If the following problems are well-addressed, this reviewer believes that the essential contribution of this paper is important for biocompatible materials.
- Small sample size of only 15 rats and the fact that it was conducted in a murine model rather than human subjects.
This study was made in pandemic conditions, the disponibility of materials and hours for work was limited in the laboratories.
- The study only evaluated short-term outcomes at 15, 30, and 45 days, so it is unclear whether the observed results would be sustained over a longer period.
The objective in the study was evaluate inflammation infiltrate and begin of dentinal bridge. In tne next study we will considerate more over a longer period evaluation.
- The study did not compare the efficacy of the three materials in terms of clinical outcomes such as pain relief or the need for additional treatments, which may be important factors in determining their usefulness in dental practice.
In according with this topic is not considered in this study, with this preliminary results in the next stage we will analyze more variables.

Reviewer 2 Report
Dear Authors
It is an honor for me to be a reviewer of your scientific work. Research topics are close to my heart and I have read with interest the assumptions and results of your research.
While studying your manuscript, I received comments, some of which are of lower importance, but some seem to me to be fundamental. Due to the variously assessed importance of my comments, let me list them not chronologically, but according to their importance.
1. Is the rodent model appropriate for dentine regeneration studies as the physiology of rodent incisors differs significantly from the physiology of human incisors? The regenerative potential between these species also seems to be significantly different, which may significantly affect the results of pulp healing. You discuss this issue in the discussion. However, you base your decision on this risky approximation on just one publication. The issue of choosing the right animal model is of significant translational importance for research.
2. Why did you choose the Wistar rat breed, males and weight 200-250g?
3. Was the age of the rats relevant to the timing of launching and terminating the experiment in relation to developmental status of the rats?
4. What is the teeth pattern in your model? Is it the same as in the humans? What is the state of tooth development?
5. Why did you choose 15, 30 and 45 days as research times?
6. How to calculate the size of groups of animals necessary to obtain reliable results?
7. You studied some procedures on the central lower incisors and some on the central upper incisors. Are the rat's upper and lower central incisors the same? Can the differences in anatomical structure not cause research results?
8. You only use pulp exposure in your control. This can expose the tooth to bacterial infection and is incomparable to testing when the pulp is protected against this option.
9. Did the rats eat freely when exposed to pain stimuli and infection from the exposed dental pulp of the control group?
10. Material and methods vaguely described: 15 rats divided into three groups (groups referring to the type of procedure in brackets) while the description under table 1 suggests that each rat had all 4 procedures performed (3 research and 1 control).
11. Page 10 line 2 you use the term Group 1 but you don't describe the groups with numbers beforehand. The reader does not know what group 1 is or if there are any other groups named by numbers.
12. Figure 2 and 3 No magnification indicated. I get the impression that the photos are at different magnifications. The images also show very different sections of the explants and it is difficult for the reader to compare them. This is difficult both within the same treatment at different times and at the same time between treatments. I suggest choosing more representative and convincing photos.
13. Figure 4.5.6 shows no differences between the groups. So study groups are also no different from controls? So can it be concluded that the use of research materials does not change anything compared to spontaneous healing?
14. Descriptions of research materials have strong and defined messages, while you support his statements only with single, often non-English literature items or even just a case report.
15. You do not describe how the sample was assessed - by one or many researchers -and whether you protected yourself from an assessment error in this regard
16. You use the name "pathological samples" to describe the explants (page 4 line 3). It seems to be not correct name.
17. There is no need to discuss prices and shelf life (page 11, lines 43-47) because the content of the study does not apply.
18. The last part of the introduction goes to the MDPI template fragment and must be removed.
Best regards
Author Response
- Is the rodent model appropriate for dentine regeneration studies as the physiology of rodent incisors differs significantly from the physiology of human incisors? The regenerative potential between these species also seems to be significantly different, which may significantly affect the results of pulp healing.You discuss this issue in the discussion. However, you base your decision on this risky approximation on just one publication. The issue of choosing the right animal model is of significant translational importance for research.
The wistar rat is one of the most popular strains used for laboratory research, so this strain serves as a model organism for the analysis of a significant number of characteristics in the biomedical and toxicological area and is considered as an important tool for research in conditions that affect human and these can be simulated in rodent model.
- Why did you choose the Wistar rat breed, males and weight 200-250g?
To have greater control in variables according with recommendation of veterinarian.
- Was the age of the rats relevant to the timing of launching and terminating the experiment in relation to developmental status of the rats?
No
- What is the teeth pattern in your model? Is it the same as in the humans? What is the state of tooth development?
Because initial formation in dentinal bridge is after 21 days.
- Why did you choose 15, 30 and 45 days as research times?
This study was made in pandemic conditions, the disponibility of materials and hours for work was limited in the laboratories.
- How to calculate the size of groups of animals necessary to obtain reliable results?
Non-probabilistic sampling for convenience, where quintuplicate sampling was carried out for each group
- You studied some procedures on the central lower incisors and some on the central upper incisors. Are the rat's upper and lower central incisors the same? Can the differences in anatomical structure not cause research results?
In the rat model dental grown is constant, since the upper and lower teeth are similar, it is the best tooth to avoid losing the evaluation area with dental grown.
- You only use pulp exposure in your control. This can expose the tooth to bacterial infection and is incomparable to testing when the pulp is protected against this option.
No, the goal is pulp necrosis and absence of the reparation, only in control group
- Did the rats eat freely when exposed to pain stimuli and infection from the exposed dental pulp of the control group?
Yes, after one hour of surgery time. The wistar stain has reported high pain umbral.
- Material and methods vaguely described: 15 rats divided into three groups (groups referring to the type of procedure in brackets) while the description under table 1 suggests that each rat had all 4 procedures performed (3 research and 1 control).
Each rat had its own control group since a split-mouth design was carried out, only the most representative image was chosen for present results.
- Page 10 line 2 you use the term Group 1 but you don't describe the groups with numbers beforehand. The reader does not know what group 1 is or if there are any other groups named by numbers.
Ok, we attend the comment
- Figure 2 and 3 No magnification indicated. I get the impression that the photos are at different magnifications. The images also show very different sections of the explants and it is difficult for the reader to compare them. This is difficult both within the same treatment at different times and at the same time between treatments. I suggest choosing more representative and convincing photos.
|
|||||||||
|
|||||||||
- Figure 4.5.6 shows no differences between the groups. So study groups are also no different from controls? So can it be concluded that the use of research materials does not change anything compared to spontaneous healing?
No, the research materials present a better response that control group, however statistically there is not difference between them.
- Descriptions of research materials have strong and defined messages, while you support his statements only with single, often non-English literature items or even just a case report.
We will attend the recommendation
- You do not describe how the sample was assessed - by one or many researchers -and whether you protected yourself from an assessment error in this regard
The evaluation was double blind study
- You use the name "pathological samples" to describe the explants (page 4 line 3). It seems to be not correct name.
Ok, we will attend the recommendation
- There is no need to discuss prices and shelf life (page 11, lines 43-47) because the content of the study does not apply.
Ok, we will attend the recommendation
- The last part of the introduction goes to the MDPI template fragment and must be removed.
ok
- Is the rodent model appropriate for dentine regeneration studies as the physiology of rodent incisors differs significantly from the physiology of human incisors? The regenerative potential between these species also seems to be significantly different, which may significantly affect the results of pulp healing.You discuss this issue in the discussion. However, you base your decision on this risky approximation on just one publication. The issue of choosing the right animal model is of significant translational importance for research.
The wistar rat is one of the most popular strains used for laboratory research, so this strain serves as a model organism for the analysis of a significant number of characteristics in the biomedical and toxicological area and is considered as an important tool for research in conditions that affect human and these can be simulated in rodent model.
- Why did you choose the Wistar rat breed, males and weight 200-250g?
To have greater control in variables according with recommendation of veterinarian.
- Was the age of the rats relevant to the timing of launching and terminating the experiment in relation to developmental status of the rats?
No
- What is the teeth pattern in your model? Is it the same as in the humans? What is the state of tooth development?
Because initial formation in dentinal bridge is after 21 days.
- Why did you choose 15, 30 and 45 days as research times?
This study was made in pandemic conditions, the disponibility of materials and hours for work was limited in the laboratories.
- How to calculate the size of groups of animals necessary to obtain reliable results?
Non-probabilistic sampling for convenience, where quintuplicate sampling was carried out for each group
- You studied some procedures on the central lower incisors and some on the central upper incisors. Are the rat's upper and lower central incisors the same? Can the differences in anatomical structure not cause research results?
In the rat model dental grown is constant, since the upper and lower teeth are similar, it is the best tooth to avoid losing the evaluation area with dental grown.
- You only use pulp exposure in your control. This can expose the tooth to bacterial infection and is incomparable to testing when the pulp is protected against this option.
No, the goal is pulp necrosis and absence of the reparation, only in control group
- Did the rats eat freely when exposed to pain stimuli and infection from the exposed dental pulp of the control group?
Yes, after one hour of surgery time. The wistar stain has reported high pain umbral.
- Material and methods vaguely described: 15 rats divided into three groups (groups referring to the type of procedure in brackets) while the description under table 1 suggests that each rat had all 4 procedures performed (3 research and 1 control).
Each rat had its own control group since a split-mouth design was carried out, only the most representative image was chosen for present results.
- Page 10 line 2 you use the term Group 1 but you don't describe the groups with numbers beforehand. The reader does not know what group 1 is or if there are any other groups named by numbers.
Ok, we attend the comment
- Figure 2 and 3 No magnification indicated. I get the impression that the photos are at different magnifications. The images also show very different sections of the explants and it is difficult for the reader to compare them. This is difficult both within the same treatment at different times and at the same time between treatments. I suggest choosing more representative and convincing photos.
|
|||||||||
|
|||||||||
- Figure 4.5.6 shows no differences between the groups. So study groups are also no different from controls? So can it be concluded that the use of research materials does not change anything compared to spontaneous healing?
No, the research materials present a better response that control group, however statistically there is not difference between them.
- Descriptions of research materials have strong and defined messages, while you support his statements only with single, often non-English literature items or even just a case report.
We will attend the recommendation
- You do not describe how the sample was assessed - by one or many researchers -and whether you protected yourself from an assessment error in this regard
The evaluation was double blind study
- You use the name "pathological samples" to describe the explants (page 4 line 3). It seems to be not correct name.
Ok, we will attend the recommendation
- There is no need to discuss prices and shelf life (page 11, lines 43-47) because the content of the study does not apply.
Ok, we will attend the recommendation
- The last part of the introduction goes to the MDPI template fragment and must be removed.
ok

Reviewer 3 Report
In the article: “Histopathological biocompatibility evaluation of TheraCalPT™, NeoMTA®, and MTA Angelus® in a murine model”, the authors discussed about the capacity of regeneration of the dentinpulp complex in a murine model with different treatments with MTA Angelus™, NeoMTA™ and TheraCal PT™.
Overall, this manuscript results very interesting, the authors clearly explain the rational of the study and discussed the topic point by point.
However, we would like to invite the authors to clarify some minor points:
1. Please check the check punctuation and spaces;
2. Figure 1; please insert the scale bar in each image;
3. Figure 2; please insert the scale bar in each image;
4. Figure 3; please insert the scale bar in each image;
5. Figures 1,2,3; are major magnification images available? Why the choice of 4X?
6. Figure 5; why standard deviation are not visible in all graphs?
7. Figure 6; why standard deviation are not visible in all graphs?
Author Response
- Please check the check punctuation and spaces;
Ok, we attend the comment
- Figure 1; please insert the scale bar in each image;
Ok, we attend the comment
- Figure 2; please insert the scale bar in each image;
Ok, we attend the comment
- Figure 3; please insert the scale bar in each image;
Ok, we attend the comment
- Figures 1,2,3; are major magnification images available? Why the choice of 4X?
To better observe all tissue and the formation of the fibro-connective tissue peudocapsule in the materials used
- Figure 5; why standard deviation are not visible in all graphs?
Only in groups where exist a visual difference, but not a statistical difference.
- Figure 6; why standard deviation are not visible in all graphs?
Only in groups where exist a visual difference, but not a statistical difference.

Round 2
Reviewer 1 Report
Accept